# Temporal transcriptomic profiling of human three-dimensional neuromuscular co-cultures

Neha Jadhav Giridhar[1,*], Bita Hambrecht[1,2,*], Maren Schenke[3,4], Bettina Seeger[3], Thorsten Bischler[5], Michael Briese[1] and Patrick Lüningschrör[1,‡]

## ABSTRACT

The principal organization of mammalian neuromuscular junctions (NMJs) shares essential features across species. However, human NMJs (hNMJs) exhibit distinct structural and physiological properties. While recent advances in stem-cell-based systems have significantly improved *in vitro* modeling of hNMJs, the extent to which these models recapitulate *in vivo* development remains unclear. Here, we performed temporal transcriptomic analysis of human three-dimensional (3D) neuromuscular co-cultures, composed of iPSC-derived motoneurons and skeletal muscle engineered from primary myoblasts. We found that the expression pattern follows a temporally coordinated gene expression program underlying NMJ maturation. The model recapitulates transcriptional features of NMJ development, including early myoblast fusion and presynaptic development, followed by a late-stage upregulation of postsynaptic markers and embryonic AChR subunits. Importantly, comparable transcriptional dynamics across two independent hiPSC lines confirm the reproducibility and robustness of this system. This study confirms on a transcriptional level that human 3D neuromuscular co-cultures are a robust and physiologically relevant model for investigating hNMJ development and function.

KEY WORDS: Human neuromuscular junction (hNMJ), 3D neuromuscular co-culture, iPSC-derived motoneurons, Temporal transcriptomics, Neuromuscular junction (NMJ) development, Skeletal muscle engineering

## INTRODUCTION

The neuromuscular junction (NMJ) is a specialized synapse that facilitates the transmission of neural signals from spinal motoneurons (MNs) to skeletal muscle fibers (Sanes and Lichtman, 1999). Although the principal organization of vertebrate/mammalian NMJs shares essential features, there are considerable interspecies

differences in the NMJ morphology. Human NMJs (hNMJs) exhibit distinctive morphological and molecular features, notably smaller presynaptic terminals and quantal vesicle size relative to muscle fiber diameter, combined with deeper postsynaptic infoldings that enlarge synaptic surface area (Slater, 2017). In addition, their synaptic proteome composition is distinct, with divergent localization of active zone proteins such as SNAP-25 (Jones et al., 2017). These morphological and proteomic differences from commonly used rodent models underscore the necessity for dedicated hNMJ systems.

Over the past decade, the neuromuscular modeling field has evolved from two-dimensional (2D) MN–myotube assays to complex three-dimensional (3D) constructs that more closely mimic native tissue architecture. Early 2D co-cultures provided valuable insights into synapse formation. They were used to model conditions like Duchenne muscular dystrophy (Maffioletti et al., 2018), myasthenia gravis (Steinbeck et al., 2016), and amyotrophic lateral sclerosis (ALS) (Stoklund Dittlau et al., 2021; Bademosi et al., 2023; Guo et al., 2020), but they lack the mechanical cues and cell–matrix interactions of muscle *in vivo*. 3D co-cultures address these limitations by embedding iPSC-derived MNs and primary human myoblast–derived fibers in a supportive matrix, enabling the formation of aligned myofibers, spontaneous contraction, and functional NMJs (Afshar Bakooshli et al., 2019; Massih et al., 2023).

Fully isogenic organoid models grown entirely from iPSCs offer the added benefit of synchronized development and inclusion of accessory cells such as Schwann cells (SCs) and endothelial populations (Pereira et al., 2021; Faustino Martins et al., 2020), but they can suffer from variability in size, necrotic centers, and lengthy, complex differentiation protocols (Leng et al., 2023; Yang et al., 2025). Fused 'assembloids' combine separate muscle and neural organoids to enhance synaptic connectivity and coordinated contractions, yet they remain challenging to scale and reproduce consistently (Yang et al., 2025). More recently, biohybrid spheroids, incorporating conductive materials and microvascular networks have demonstrated real-time electrophysiological readouts in ALS disease modeling (Shin et al., 2025). Despite these advances, a systematic comparison of how closely each system reflects hNMJ morphology, function, and developmental timing is still lacking.

In this study, we used a 3D human neuromuscular co-culture system that combines iPSC-derived spinal MNs with primary human myoblast-derived muscle fibers embedded in a geltrex-fibrin-based hydrogel matrix within dumbbell-shaped PDMS molds. This configuration promotes alignment, myotube fusion, contractility, and NMJ formation. While functional assays were not performed here, prior studies using nearly identical co-culture conditions have demonstrated spontaneous contractions and α-bungarotoxin-positive NMJs (Afshar Bakooshli et al., 2019; Massih et al., 2023). We selected this platform for its reproducibility, accessibility, and compatibility with live imaging and defined time molecular profiling.

[1]Institute of Clinical Neurobiology, University Hospital Würzburg, Versbacher Str. 5, 97078 Würzburg, Germany. [2]Institute for Pathophysiology and Allergy Research, Center for Pathophysiology, Infectiology and Immunology, Medical University of Vienna Währinger Gürtel 18-20, 1090 Vienna, Austria. [3]Institute for Food Quality and Safety, Research Group Food Toxicology and Alternative/Complementary Methods to Animal Experiments, University of Veterinary Medicine Hannover, 30559 Hannover, Germany. [4]Bielefeld University, Medical School OWL, Anatomy and Cell Biology, 33615 Bielefeld, Germany. [5]Core Unit Systems Medicine, University of Würzburg, D-97080 Würzburg, Germany.
*These authors contributed equally to this work

‡Author for correspondence (Lueningsch_P@ukw.de)

N.J.G., 0000-0001-6544-4102; B.H., 0009-0005-7908-2305; P.L., 0000-0002-3883-6554

To better understand the molecular events underlying NMJ formation, we conducted a temporal transcriptomic analysis of human 3D neuromuscular co-cultures. Our data revealed a clear temporal trajectory of NMJ formation, beginning with transcriptional changes indicative of myoblast fusion, followed by upregulation of genes associated with presynaptic and postsynaptic assembly. This was succeeded by a distinct induction of transcripts involved in cholinergic signaling and muscle contraction. Together, these data demonstrate that human 3D neuromuscular co-cultures recapitulate transcriptional features of embryonic NMJ development and provide a temporal reference framework to inform future studies of NMJ formation and disease.

## RESULTS
### Temporal transcriptomic profiling of human neuromuscular co-cultures reveals cellular interactions and gene expression dynamics
Neuromuscular co-cultures were generated as previously described (Massih et al., 2023). Our prior work using healthy iPSC lines showed that axon outgrowth in neuromuscular co-cultures peaked after 3 weeks, coinciding with NMJ formation and muscle function (Massih et al., 2023). MN-triggered muscle contraction remained stable for up to 6 weeks, although neurite outgrowth plateaued or slightly declined beyond 3 weeks. Later time points have not been investigated so far. Therefore, we chose 1, 3, and 6 weeks as time points for bulk RNA sequencing (RNA-seq) (Fig. 1A).

To analyze dynamic changes in gene expression over time, we first identified transcripts that were differentially expressed between week 3 and week 1 (Table S1), and separately those that were differentially expressed between week 6 and week 3 (Table S2). From this analysis, 2837 transcripts were found to be differentially expressed in both comparisons (Fig. 1B,C; Table S3). Additionally, 6415 transcripts were uniquely regulated between weeks 1 and 3 (Fig. 1D), with no significant change from week 3 to week 6. Among these, 5071 were upregulated and 1344 were downregulated (Fig. 1F, groups 1 and 4). Conversely, 2595 transcripts were uniquely regulated between weeks 3 and 6, with stable expression from week 1 to 3 (Fig. 1E). Of these, 1071 were upregulated and 1524 downregulated (Fig. 1F, groups 2 and 5).

Based on these temporal transcriptional dynamics (Fig. 1C-E), we classified the differently expressed transcripts into eight groups (Fig. 1F). The first group consists of transcripts significantly upregulated from 1 to 3 weeks that were stably expressed from 3 to 6 weeks and represents the most prominent group, containing 5071 transcripts. Among these, we detected *TUBA1B*, *ACTB*, and *FILIP1* encoding Tubulin alpha-1B chain, beta-Actin, and filamin A interacting protein 1, a structural protein involved in neuronal and muscle function and integrity (Fig. 1G) (Roos et al., 2023).

The second group contains transcripts stably expressed from 1 to 3 weeks and upregulated from 3 to 6 weeks. In this group, *AGT* was the transcript with the strongest upregulation (Fig. 1H). It encodes angiotensinogen, a precursor to angiotensin, which is secreted by skeletal muscle as part of the local renin-angiotensin system that regulates metabolism, regeneration, and inflammation (Gorman et al., 2014; Powers et al., 2018). We also detected *ACTN2*, encoding the Z disc protein Actinin 2, essential for muscle development and function (Mancinelli et al., 2021).

Group 3 is defined by transcripts upregulated during both time windows, such as *NTRK2*, *MGP*, *CXCL12*, and *SPON1* (Fig. 1I). The neurotrophin receptor TrkB, encoded by *NTRK2*, is well established for its roles in neuronal survival, differentiation, synaptic transmission, and plasticity (Andreska et al., 2020). It

also supports muscle development and regeneration, with muscle-derived BDNF activating TrkB via autocrine and paracrine signaling to promote fiber differentiation and repair (Mancinelli et al., 2021; Chevrel et al., 2006). *MGP* encoding the Matrix gla protein, has been implicated in modulating the muscle development program by regulating myostatin (Ahmad et al., 2017). *SPON1 and CXCL12*, encode proteins involved in axon guidance and regeneration (Schubert et al., 2006; Negro et al., 2017).

Group 4 includes transcripts that were significantly downregulated from 1 to 3 weeks but remained stably expressed thereafter. Among these, we detected *PHLDA1*, *CITED2*, and *IFRD1*, which encode proteins involved in regulating NF-kappa B signaling and inflammation (Fig. 1J) (Lou et al., 2011; Micheli et al., 2011; Peng et al., 2022).

Group 5 includes transcripts like *MMP1* and *MMP3*, which are stably expressed early and downregulated from 3 to 6 weeks (Fig. 1K). Matrix Metalloprotease-1 (MMP1) degrades collagen, while Matrix Metalloprotease-3 (MMP3) targets various ECM components, including collagen, fibronectin, and laminin (Cabral-Pacheco et al., 2020). Both enzymes are key players in ECM remodeling, tissue repair, inflammation, and have specific roles in NMJ organization and maintenance (Wan et al., 2021; Dear et al., 2016; Bloch-Gallego, 2015; VanSaun et al., 2003).

Group 6 contains transcripts that were downregulated during both time windows. It represents the smallest group with only 135 transcripts, such as *CXCL8* and *IL24* (Fig. 1L).

In group 7, we identified transcripts upregulated from week 1 to 3 and downregulated from week 3 to 6. Among these, we detected *CCL7*, *CCL8*, and *CCL11*, suggesting a transit function for these chemokines in neuromuscular development (Fig. 1M) (Boss-Kennedy et al., 2024; Blanc et al., 2020).

Group 8 is the second-smallest group, consisting of transcripts transiently downregulated. Among these, we detected *miR-3648-1* and *miR-3648-2*, human-specific microRNA clusters that might be interesting candidates for functional analysis in future studies (Fig. 1N).

### Gene ontology (GO)-term analysis of differentially expressed transcripts in neuromuscular co-cultures
We performed GO analysis as an unbiased approach to determine the function of the differentially expressed transcripts in these eight groups.

GO term analysis of transcripts upregulated initially and stably expressed later revealed the enrichment of transcripts, which function in regulating synaptic transmission, cell adhesion, and ion transport (Fig. 2A; Table S4).

Transcripts, specifically upregulated from 3 to 6 weeks, were mainly enriched for functions in muscle contraction, antigen presentation, and acetylcholine receptor function (Fig. 2A; Table S5).

GO term analysis of transcripts upregulated during both periods revealed the enrichment of transcripts functioning in cell adhesion and synapse development (Fig. 2A; Table S6). GO term analysis of transcripts that were initially downregulated but remained unchanged later revealed strong enrichment in RNA processing, including more than 60 small nucleolar RNAs (snoRNAs) (Fig. 2B; Table S7). Interestingly, both snoRNAs and small nuclear RNAs (snRNAs) have been implicated in myogenesis, as well as skeletal muscle development and regeneration (Kanakis et al., 2021). Their downregulation may indicate a shift in ribosome biogenesis or function during NMJ maturation, potentially supporting specialized translation programs.

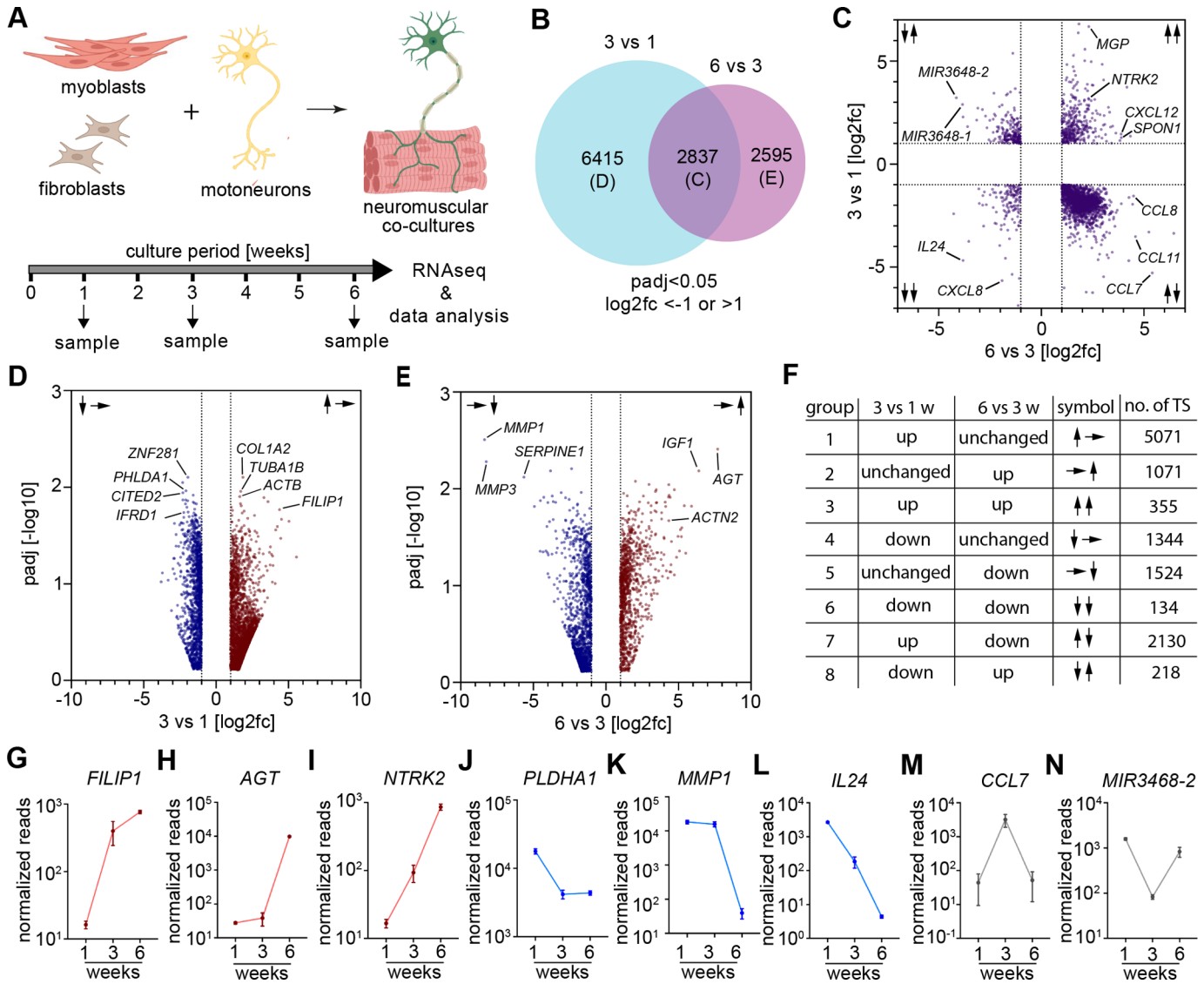

**Fig. 1.** Temporal transcriptomic analysis of human neuromuscular co-cultures reveals dynamic gene expression patterns associated with NMJ development. (A) Schematic overview of the experimental design for temporal transcriptomic profiling at 1, 3, and 6 weeks of co-culture. (B) Venn diagram illustrating the overlap of differentially expressed transcripts between the 3 week versus 1 week and 6 week versus 3 week comparisons. The letters indicate the figure in which the respective transcripts are shown in more detail. (C) Scatterplot showing the magnitude of change (fold change, log2) of the transcripts significantly altered in 3 week versus 1 week and 6 week versus 3 week co-cultures. $n$=3 biological replicates. Arrows indicate gene expression changes across two comparisons: ↑↑ (top right): upregulated in both 3 weeks versus 1 week and 6 weeks versus 3 weeks. ↓↓ (bottom left): downregulated in both. ↑↓ (top left): upregulated from 1 week to 3 weeks, then downregulated from 3 weeks to 6 weeks. ↓↑ (bottom right): downregulated from 1 week to 3 weeks, then upregulated from 3 weeks to 6 weeks. $n$=3 biological replicates. (D,E) Volcano plots highlighting significantly regulated genes between 1 week and 3 weeks (D), and 3 weeks and 6 weeks (E). Arrows indicate gene expression changes: ↓→ (left, D): downregulated from 1 week to 3 weeks, unchanged to 6 weeks. ↑→ (right, D): upregulated from 1 week to 3 weeks, unchanged to 6 weeks. →↓ (left, E): stable from 1 week to 3 weeks, then downregulated to 6 weeks. →↑ (right, E): stable from 1 week to 3 weeks, then upregulated to 6 weeks. (F) Summary table showing eight distinct temporal expression patterns. (G–N) Expression pattern of representative transcripts from each of the eight groups. All data are shown as mean±s.d. $n$=3 biological replicates.

Next, we found an enrichment of transcripts that function in immune response in the groups of transcripts specifically downregulated from 3 to 6 weeks and downregulated during both periods (Fig. 2B; Tables S8-S9). This downregulation of immune-related transcripts likely reflects a resolution of early stress or inflammatory responses triggered by initial co-culture conditions, such as matrix remodeling or cellular integration. It may also indicate a shift toward a more stable and mature neuromuscular environment, supporting synaptic refinement and sustained NMJ functionality over time.

Interestingly, the group of transcripts upregulated early and downregulated later was enriched for transcripts associated with functions in mitosis (Fig. 2C; Table S10). This pattern likely reflects

that myoblasts and fibroblasts in cell culture become post-mitotic with ongoing culturing.

Overall, we classified the most enriched biological processes into three principal phases that mirror the established *in vivo* sequence of hNMJ development. During the early myogenesis phase, the enrichment of terms such as cell division, mitotic cell cycle, chromosome segregation, mitotic spindle assembly, mitotic cytokinesis, extracellular matrix (ECM) organization, signal transduction, and cAMP-mediated signaling underscores myoblast proliferation, ECM remodeling, and readiness for fusion. The synapse assembly phase encompasses GO terms including synapse assembly, chemical synaptic transmission, acetylcholine receptor signaling,

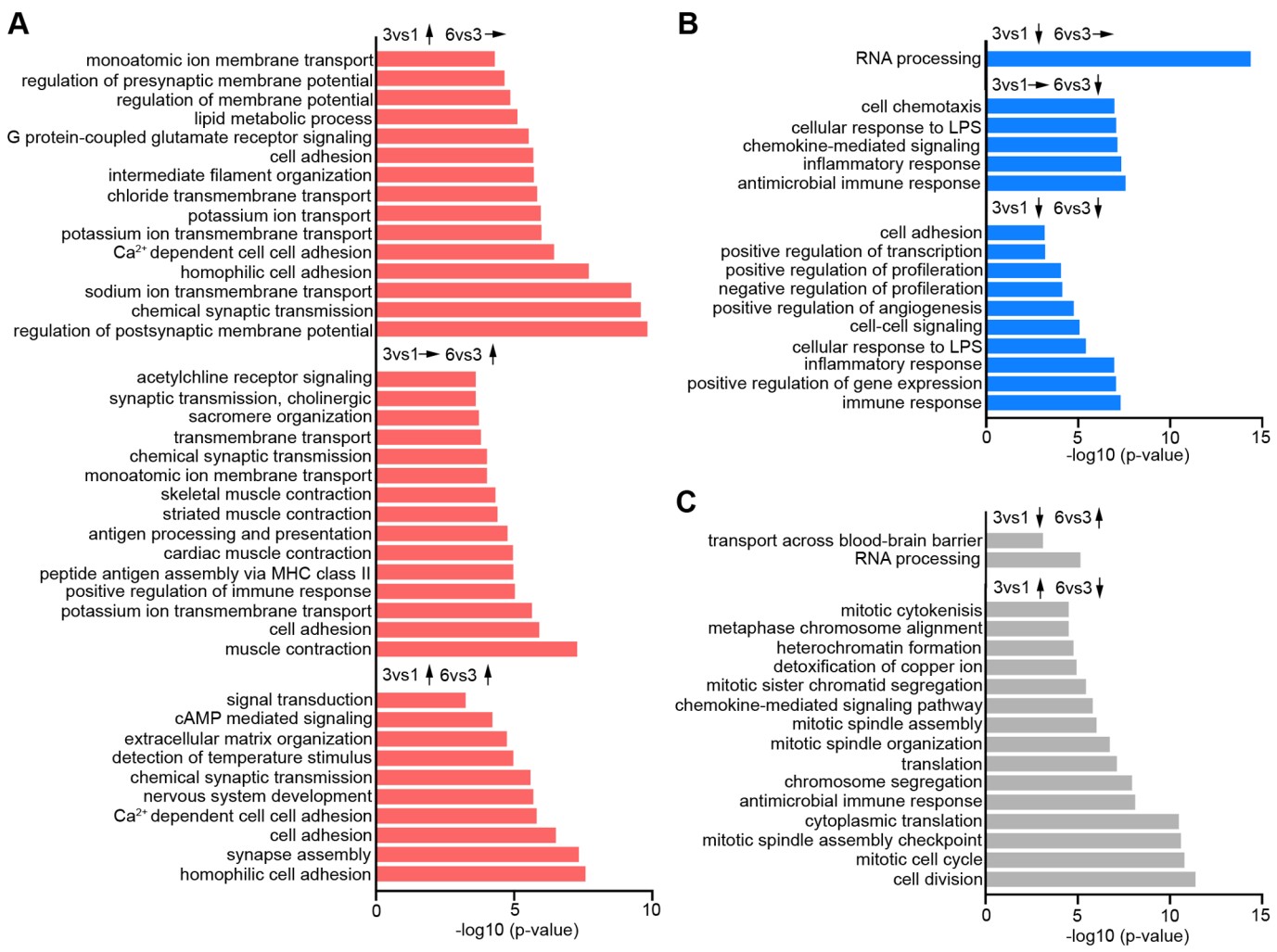

**Fig. 2. GO analysis of temporally expressed transcripts across developmental time points.** (A-C) Enriched GO biological processes for genes significantly upregulated (A), downregulated (B), and non-differentially expressed (C) across both the 3 week versus 1 week and 6 week versus 3 week comparisons. Arrows indicate the directionality and stability of gene expression over time. The top 15 GO terms that include six or more genes and a *P*-value<0.001 are shown.

cholinergic synaptic transmission, regulation of postsynaptic membrane potential, cell–cell adhesion, homophilic cell adhesion, GPCR-coupled glutamate receptor signaling, cell–cell signaling, and cell adhesion, highlighting the formation of both pre- and post-synaptic structures and adhesion machinery. In the functional maturation phase, enriched processes such as sodium/potassium/chloride ion transport, Ca²-dependent cell–cell adhesion, regulation of membrane potential, monoatomic ion membrane transport, lipid metabolic process, sarcomere organization/striated muscle contraction/muscle contraction reflect the development of electrical excitability, ion handling, and contractile capability. These three phases – early myogenesis, synapse assembly, and functional maturation – thereby offer a streamlined, stage-focused framework for interpreting the time-dependent shifts in our 3D neuromuscular co-cultures.

### Differentially expressed transcripts critical for muscle and NMJ formation

Next, we focused our analysis on transcripts involved in different aspects of cell adhesion, as well as muscle and NMJ maturation. First, we analyzed the expression of cell adhesion molecules

(CAMs) involved in myoblast fusion, a critical step in skeletal muscle formation following myogenic commitment and differentiation. This process is closely linked to the upregulation of muscle-specific transcription factors such as MyoD, which we observed increasing from weeks 3 to 6, consistent with its role in promoting cell cycle exit and fusion (Redfield et al., 1997). Myoblast fusion proceeds through a coordinated sequence of events involving classical cadherins (M-, N- and R-cadherin), NCAM, integrins, and other CAMs, including VCAM-1 and JAMs, which mediate cell recognition, adhesion, and cytoskeletal remodeling (Przewozniak et al., 2013; Charrasse et al., 2006). In line with these mechanisms, we detected upregulation of the transcripts for CDH15 (M-cadherin), CDH2 (N-cadherin), NCAM1, and VCAM1 from weeks 3 to 6, indicating activation of adhesion pathways essential for myoblast fusion (Fig. 3A).

Protocadherins (Pcdhs), the most diverse subgroup within the cadherin superfamily, are key mediators of cell–cell interactions and play critical roles in neural circuit formation, including dendritic arborization, axon guidance, synaptogenesis, and synapse elimination (Mancinelli et al., 2021). Notably, the Pcdh-γ cluster is essential for MN survival and spinal cord patterning (Wang et al.,

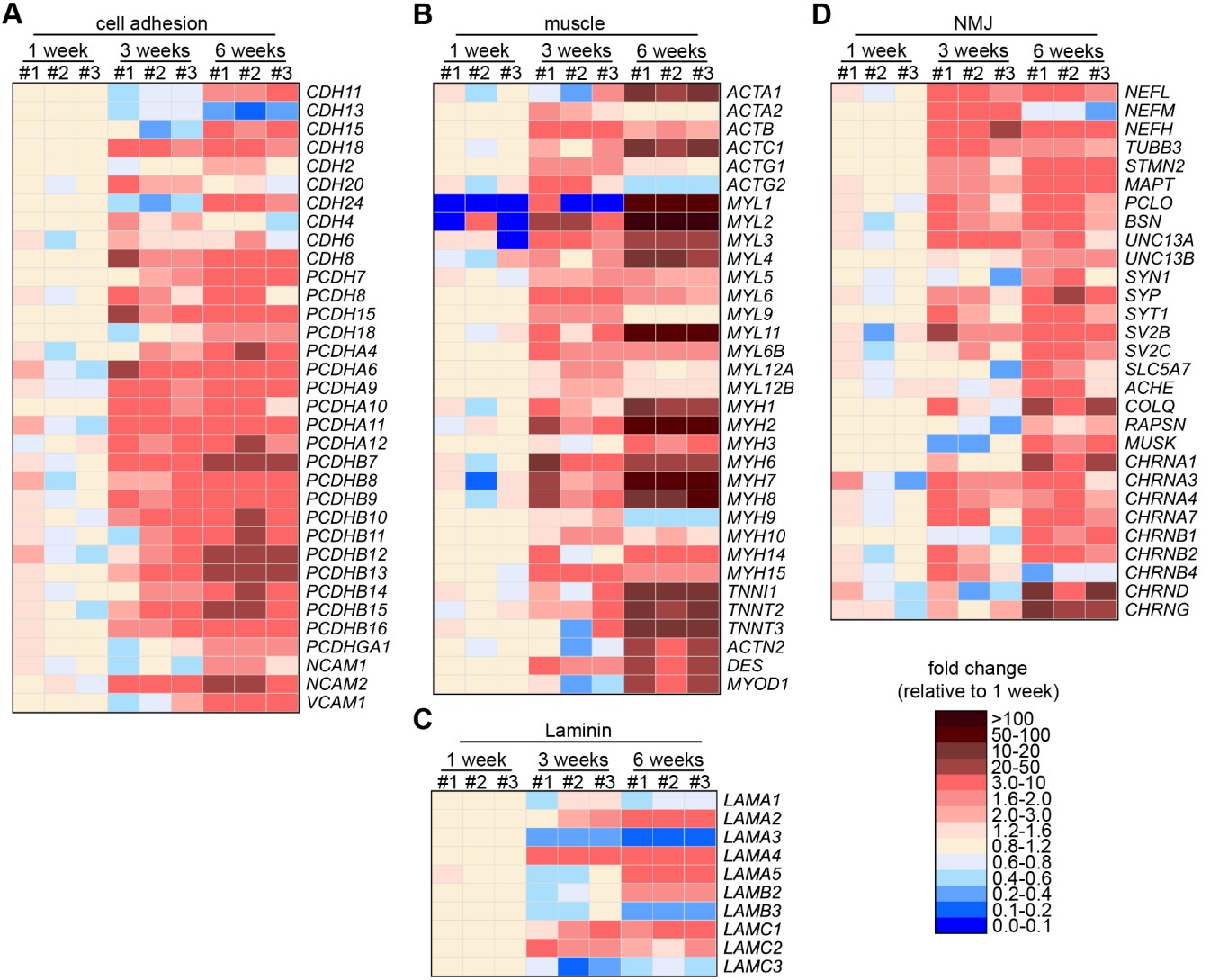

**Fig. 3. Heatmaps of differentially expressed transcripts associated with neuromuscular development.** (A-D) Heat maps showing the temporal expression of transcripts related to cell adhesion (A), muscle cytoskeleton (B), Laminin (C) and NMJ structure and function (D). The expression levels are shown as fold change of the normalized read counts adjusted to the mean levels of 1-week-old cultures.

2002), and Pcdhs contribute to excitatory synapse formation through axon–dendrite interactions (Mancini et al., 2020). In our dataset, protocadherin genes were among the most highly expressed adhesion molecules, with several clustered Pcdhs (α-, β-, and γ-types) upregulated over time. In particular, *PCDHB7*, *PCDHB12*, and PCDHB13 showed marked upregulation between weeks 3 and 6, suggesting a potential role in NMJ maturation.

Previously, we observed the initiation of muscle function within three weeks; however, transcriptomic analysis indicates a continued maturation of muscle tissue beyond this point. Notably, we observed sustained upregulation of several myosin light chains (Fig. 3B). In addition, key Z-disc structural components, including *ACTN2* (α-actinin-2), which cross-link actin filaments and anchors them at the Z-disk, and *DES* (DESMIN), an intermediate filament protein essential for maintaining muscle fiber integrity, were also upregulated (Fig. 3B). Furthermore, expression of *TNNI1*, *TNNI2*, and *TNNI3* – encoding the slow skeletal, fast skeletal, and cardiac isoforms of troponin I, respectively – was elevated, consistent with ongoing sarcomeric organization and muscle cell differentiation (Abdul-Hussein et al., 2012; Ravenscroft et al., 2018). Collectively,

these transcriptional changes indicate progressive maturation of muscle architecture and function.

Laminins, key components of the basal lamina, are heterotrimeric glycoproteins composed of α, β, and γ chains and are critical for myogenesis and synaptogenesis. Notably, laminin isoforms containing α4, α5, and β2 chains are enriched at NMJs, where they contribute to NMJ stability and the organization of presynaptic active zones by interacting with receptors such as integrins, dystroglycan, and voltage-gated calcium channels (Rogers and Nishimune, 2017). Our data revealed dynamic changes in laminin chain expression during NMJ development. *LAMA1*, *LAMA3*, *LAMB3*, and *LAMC3* were significantly downregulated during the first 3 weeks, while *LAMA2 and LAMA4* were upregulated in the same period. From weeks 3 to 6, *LAMA5* and *LAMB2* transcripts showed increased expression as previously reported (Rogers and Nishimune, 2017) (Fig. 3C). These temporal shifts in laminin subunit expression align with their known enrichment at the NMJ and suggest stage-specific remodeling of the ECM to support synaptic maturation.

In agreement with our previous observation on axon growth decreasing after 3 weeks, we found a marked upregulation of

transcripts encoding axonal cytoskeleton proteins (*STMN2*, *MAPT*, *TUBB3*, *NEFL*, *NEFH*) from 1 to 3 weeks. Correlating with the maturation of the presynaptic compartment, we found an early upregulation of transcripts encoding for synaptic vesicle proteins such as *SV2B*, *SV2C*, *SYN1*, *SYP*, and *SYT* and the presynaptic matrix proteins *PCLO* and *BSN*. However, transcripts such as *RAPSN* and *MUSK*, which encode postsynaptic components and transcripts involved in acetylcholine metabolism (*COLQ*, *ACHE*, *SLC5A7*), were upregulated late (Fig. 3D). In summary, these data support the notion that the maturation of the presynaptic compartment precedes NMJ formation and function. Consistent with findings in mice, postsynaptic maturation at the NMJ occurs slightly later than or in parallel with presynaptic development but lags in reaching full morphological complexity (Sanes and Lichtman, 1999).

During early postnatal development, the mammalian NMJ undergoes molecular and structural changes essential for functional maturation. Initially, embryonic AChRs (α2βγδ) support pre-patterned receptor clusters, with the γ-subunit critical for early synaptic organization (Liu et al., 2008). As innervation stabilizes, MN-derived signals activate synapse-specific transcription in subsynaptic myonuclei, repressing the γ-subunit and inducing ε-subunit gene expression (Sanes and Lichtman, 1999). This γ-to-ε switch generates the adult AChR form (α2βδε), altering channel properties (Witzemann et al., 1996). We analyzed AChR subunit expression and found marked upregulation of AChR transcripts. From weeks 3 to 6, CHRND and CHRNG were strongly induced (Fig. 3D), while CHRNE remained unchanged. This suggests that NMJs in these co-cultures resemble an embryonic stage. In contrast, a previous study reported CHRNE upregulation in 3D neuromuscular co-cultures after 2 weeks (Afshar Bakooshli et al., 2019). However, spatial expression was not assessed. Our findings suggest extended culture duration may be needed for full maturation. Including SCs may enhance this, as terminal SCs (tSCs) provide signals regulating axonal growth, synaptic organization, and neurotransmission.

In summary, transcriptomic profiling revealed temporally coordinated expression changes underlying NMJ maturation. Early presynaptic development was marked by upregulation of axonal cytoskeleton and synaptic vesicle genes, while postsynaptic markers such as *RAPSN*, *MUSK*, and AChR subunits *CHRND* and *CHRNG* were induced later, indicating embryonic-stage NMJ formation. Key regulators of myoblast fusion and muscle structural genes were progressively upregulated, alongside stage-specific laminin isoform expression, supporting continued maturation of muscle architecture and synaptic specialization from weeks 3 to 6.

### Comparable temporal transcriptional dynamics in hMNs from distinct hiPSC lines

Due to the significant phenotypic heterogeneity between distinct hiPSC lines, reproducing key findings and integrating data across different research groups is often difficult. Thus, we validated our findings using MNs derived from a second hiPSC line (iPSC#2). Principal component analysis (PCA) revealed that PC1 distinguishes between the different time points with an obvious progression from 1 to 6 weeks, regardless of the hiPSC line used to generate the co-cultures. PC2 distinguishes between co-cultures generated from different hiPSCs (Fig. 4A).

Hierarchical clustering of RNA-seq samples based on Euclidean distance showed that gene expression patterns were primarily organized by time points in culture rather than by iPSC line (Fig. 4B). Samples collected at 1, 3, and 6 weeks each formed distinct clusters, highlighting developmental stage as the dominant

factor shaping transcriptional profiles. As expected, week 1 samples from both iPSC#1 and iPSC#2 were grouped together and clearly separated from those at later stages. Week 3 and 6 samples also clustered by time point, with greater similarity between them compared to week 1, reflecting progressive maturation of the co-cultures. Within each time point, samples were further segregated according to iPSC line, suggesting a subtle but consistent line-specific transcriptional signature. This trend was most evident at 6 weeks, where iPSC#2 samples clustered more tightly, indicating higher within-line consistency.

Overall, these findings indicate that both iPSC lines follow a comparable temporal gene expression trajectory, with developmental stage driving the major transcriptomic changes. However, line-specific variation remains detectable, particularly at later stages, likely reflecting inherent biological differences between donor lines rather than delayed or divergent differentiation.

To further analyze the differently expressed transcripts in the neuromuscular co-cultures generated with both hiPSC lines, we examined the significantly expressed transcripts after 3 weeks compared to 1 week (Fig. 4C). This comparison revealed a significant correlation between both hiPSC lines. We observed a similar trend after examining the significantly expressed transcripts after 6 weeks compared to 3 weeks, revealing a strong correlation between both hiPSC lines (Fig. 4D). In summary, our analysis demonstrates that hMNs derived from two independent hiPSC lines and two different differentiation protocols exhibit highly comparable transcriptional profiles over time in neuromuscular co-cultures.

### DISCUSSION

While our bulk RNA-seq data delineate coordinated waves of myogenic, neuronal, and synaptic gene expression, this approach inherently averages signals across heterogeneous cell populations, thereby obscuring lineage-specific contributions. Because bulk RNA-seq does not resolve cell-type-specific expression, it precludes attribution of transcriptional changes to defined lineages. Future studies employing single-nucleus RNA-seq (snRNA-seq) could address this limitation and offer improved cellular resolution. Alternatively, RNA-FISH or immunostaining approaches could be used to provide spatial and cell-type-specific validation. However, such experiments were beyond the scope of the present study and represent valuable opportunities for future work. Although functional assays were not performed in this study, prior reports using nearly identical co-culture conditions have demonstrated spontaneous contractions and α-bungarotoxin–positive NMJs (Afshar Bakooshli et al., 2019; Massih et al., 2023), supporting the expectation that our system can achieve functional connectivity.

The absence of an early reference point (day *in vitro* 0, DIV0) is another limitation of this study, which restricts our ability to discriminate between transcriptional programs associated with initial differentiation versus those tied to later maturation. As a result, early myogenic or synaptogenic events occurring before our 1-week baseline may be underrepresented. To partially mitigate this limitation, we reorganized GO-term enrichment results into three biologically relevant stages — early myogenesis, synapse assembly, and functional maturation. This approach helps to contextualize the temporal dynamics of gene expression observed in our 3D neuromuscular co-cultures. However, imaging-based approaches combined with functional assays, such as electrophysiological recordings or contractility measurements, will be essential for validating these molecular observations.

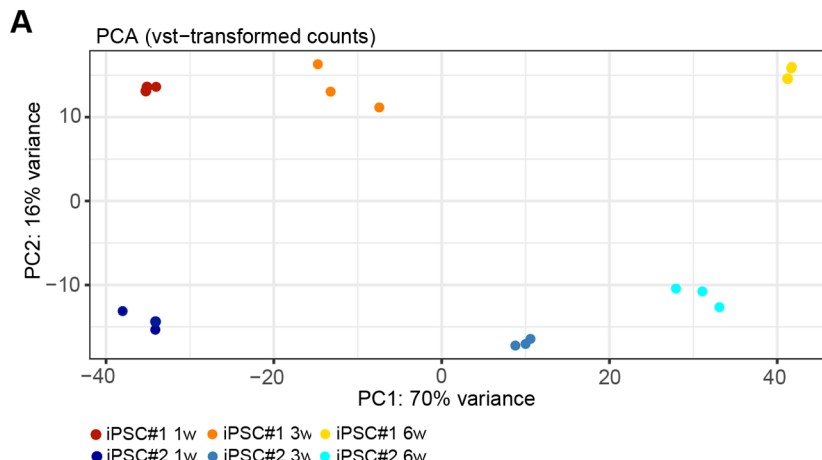

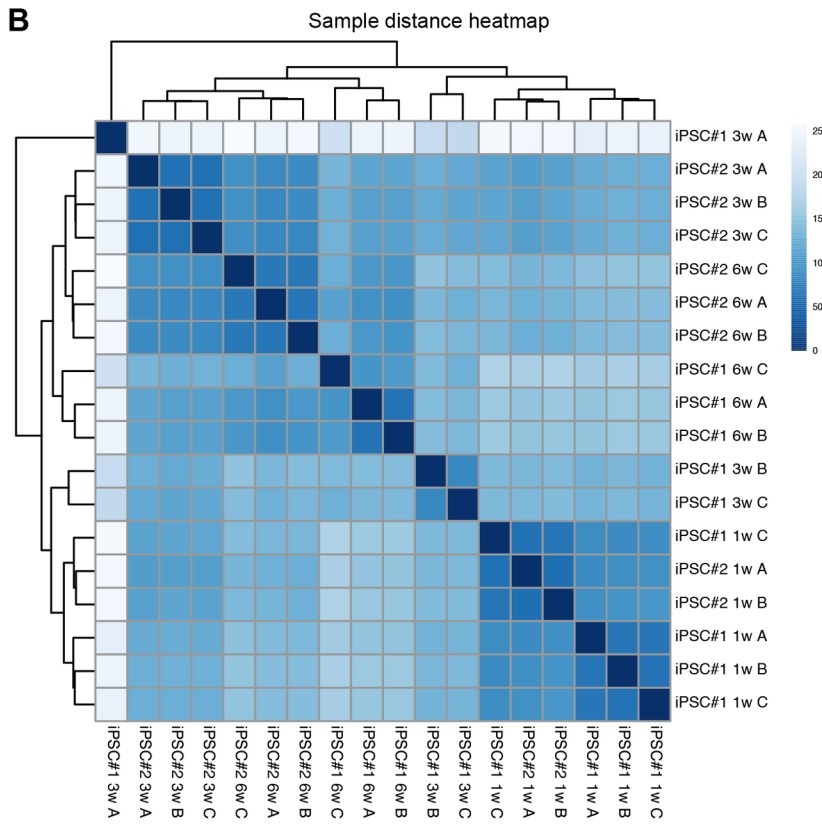

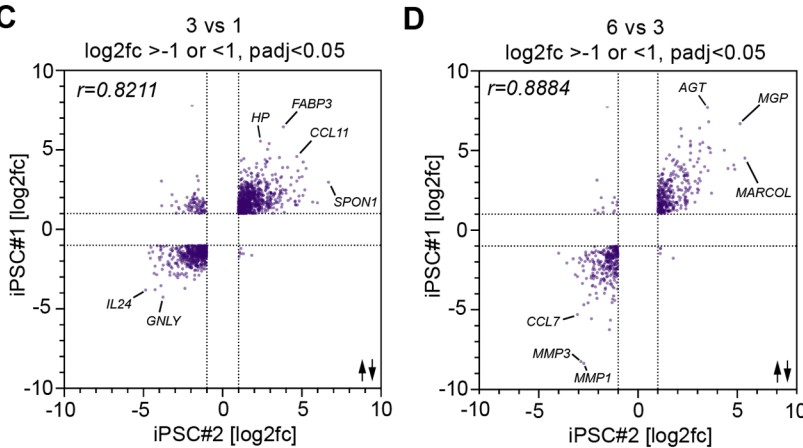

**Fig. 4. Validation of temporal transcriptomics using an independent hiPSC line.** (A) PCA of variance-stabilizing transformed (VST) gene expression shows separation by time along PC1 (70% variance) and by hiPSC line along PC2 (16% variance). (B) Sample distance heatmap based on Euclidean distances. Hierarchical clustering shows that samples primarily group by time point, with additional line-specific clustering within each stage. (C) Scatterplot of log2 fold changes in 3-week co-cultures comparing hiPSC#1 with hiPSC#2 (*n*=3); Pearson r=−0.8211. (D) Scatterplot of log2 fold changes in 6-week co-cultures comparing hiPSC#1 vs hiPSC#2 (*n*=3); Pearson r=−0.8884.

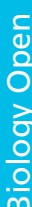

Our current focus on a human-specific co-culture model offers unique insights into hNMJ development but lacks a direct comparative analysis with rodent datasets. Such comparisons could contextualize species-specific features and serve as benchmarks for evaluating the fidelity of *in vitro* human systems. Implementing snRNA-seq or scRNA-seq across embryonic, neonatal, and adult tissues – including cell types such as PAX7[+] satellite cells, HES1[+]/TRIB1[+] progenitors, endothelial cells, and immune populations – could further delineate developmental stage-specific contributions and enhance our understanding of hNMJ maturation (Cai et al., 2023; Xu et al., 2023). In parallel, future work will also address transcript-level complexity, as alternative splicing represents an important dimension of transcriptome regulation not examined here due to library preparation and sequencing constraints. We also acknowledge that the dataset presented here is derived from only two independent iPSC lines. Expanding future analyses to include additional donor genetic backgrounds will be important to capture inter-individual variability and to strengthen the generalizability of the resulting transcriptional reference.

The use of primary human myoblasts offers a reliable platform, given their consistent differentiation into contractile myotubes within 2 weeks (Afshar Bakooshli et al., 2019; Massih et al., 2023). This contrasts with iPSC-derived myotubes, which often require extended culture durations and demonstrate variable fusion efficiency (Borchin et al., 2013; Pinton et al., 2023). While the co-culture system used here provides a simplified yet reproducible model, more complex approaches – such as neuromuscular organoids, assembloids, or biohybrid constructs – introduce greater cellular diversity and enable embedded biosensing but often encounter challenges related to variable maturation, reduced viability, and technical complexity (Leng et al., 2023; Yang et al., 2025; Shin et al., 2025).

Several strategies could advance the neuromuscular co-culture system towards a more mature 'adult-like' model. Targeted analysis of exon usage in synaptic genes such as *AGRIN* and *NEUREGULIN-1* (Kim et al., 2017; Inoue et al., 2021; Kong et al., 2023) could reveal post-transcriptional regulation that shapes NMJ development. To promote NMJ maturation, extending the culture period beyond 6 weeks may allow for the full AChR γ→ε subunit transition (Liu et al., 2008; Li et al., 2024). Inclusion of tSCs could further enhance postsynaptic specialization and synaptic stability, while Neuregulin-1 supplementation (Inoue et al., 2021; Kong et al., 2023) and matrix stabilization using protease inhibitors (Boucard et al., 2022; Ng et al., 2015; Collen et al., 1998) may improve structural and biochemical fidelity. Mechanical conditioning – via aligned micro/nanofibers, bone-mimetic scaffolds, or cyclic stretch protocols that activate ILK–PKC–ROCK signaling – has the potential to strengthen contractile force and synaptic robustness (Jafari et al., 2023; Wang et al., 2023). As muscle mechanosensitivity is highly context dependent, incorporating tunable stiffness gradients or programmable mechanical loading may also enhance reproducibility and alignment (Kelp et al., 2024).

Finally, future adaptations incorporating isogenic, patient-derived iPSC MNs and muscle cells will facilitate disease modeling, especially in contexts where developmental timing and lineage-matching are critical for accurate phenotypic manifestation. By addressing current limitations through multi-omic integration, spatial validation, functional assays, and advanced biomechanical engineering – and by benchmarking against *in vivo* and rodent models – this platform can be used for studying hNMJ development, disease, and therapeutic discovery.

## MATERIALS AND METHODS
### Cell culture
#### Myoblast and Fibroblast culture
Myoblast and Fibroblast culture was carried out as previously described (Massih et al., 2023). Briefly, Primary human skeletal muscle myoblasts (Lonza, CC-2580) were cultured in F-10 media (Gibco, 11550043) with 20% fetal bovine serum (Capricorn Scientific, FCS-62A), 5 ng/ml basic fibroblast growth factor (bFGF) (PeproTech, 100-18B), and 100 µg/ml penicillin-streptomycin (Gibco, 15-140-122), as per Afshar Bakooshli et al. (2019) with minor modifications. Cells were cultured until 80–90% confluency and used between passages 4–10. Primary human fibroblasts (Thermo Fisher Scientific, C0045C) were maintained until confluency. The medium was changed every other day for both cell types.

#### Differentiation of hiPSC line#1 into MNs
MNs derived from the hiPSC line IMR90-4 (hiPSC line#1) were originally purchased from WiCell (Yu et al., 2007), and differentiated as previously described by Schenke et al. (2020), based on Kroehne et al. (2017) with slight changes. iPSCs were differentiated into NPCs, which were then cultured for 10 passages in expansion medium with 3 µM CHIR99021 (CHIR), 0.5 µM Purmorphamine (PMA), and 150 µM ascorbic acid (AA) for three passages and with 0.5 µM smoothened agonist (SAG) (EMD Millipore, 566660) instead of 0.5 µM PMA starting in passage 4 on Matrigel-coated dishes. For MN specification, cells were cultured for 6 days in neuronal medium supplemented with 0.5 µM SAG, 1 µM RA, 1 ng/ml GDNF, and 2 ng/ml BDNF and cryopreserved.

After defrosting, for final MN maturation, the medium was switched to neuronal medium supplemented with 200 µM AA, 2 ng/ml GDNF and BDNF, 1 ng/ml transforming growth factor beta (TGFß3) (Sigma, SRP3171), 200 µM dbcAMP, and 10 µM tert-Butyl(2S)-2-[[(2S)-2-[[2-(3,5-difluorophenyl)acetyl]amino]propanoyl]-amino]-2 phenylacetate (DAPT) (Cayman Chemical Company, 13197) for at least 14 days.

#### Differentiation of hiPSC line#2 into MNs
MNs derived from the previously used hiPSC line 34D6 (iPSC-line#2) (Selvaraj et al., 2018) were differentiated according to a previously reported protocol with minor changes (Reinhardt et al., 2013). Briefly, hiPSCs were expanded in mTeSR Plus medium (Stemcell Technologies, 05825) on Matrigel-coated (1:100) (Corning, 356234) dishes. When cells reached 80–90% confluency, they were split at a 1:5 ratio using ReLeSR reagent (Stemcell Technologies, 05872). To enhance survival, ROCK inhibitor (10 µM) (StemMACS™ Y27632, Miltenyi Biotec, 130-106-538) was added for 24 h post-splitting.

For neuronal induction, mTeSR Plus was supplemented with 10 µM SB431542 (AdooQ BioScience, A10826-50), 1 µM dorsomorphin homolog 1 (DMH1) (R&D Systems, 4126), 3 µM CHIR99021 (Cayman Chemical Company, 13122), and 0.5 µM Purmorphamine (PMA) (Cayman Chemical Company, 10009634). On day 2, the medium was switched to neuronal medium supplemented with the same small molecules. The neuronal medium consisted of Neurobasal medium (Gibco, 21103049), Dulbecco's Modified Eagle's Medium F-12 (DMEM/F-12) (Gibco, 21331046), MACS NeuroBrew-21 (Miltenyi Biotech, 130-097-263), N-2 Supplement (Gibco, 17502048), and 100 µg/ml Penicillin/Streptomycin/Glutamax (Gibco, 10378016). On day 4, the medium was replaced with an expansion medium composed of neuronal medium supplemented with 3 µM CHIR99021, 0.5 µM PMA, and 150 µM Ascorbic Acid (AA) (Sigma, A92902). When cells reached 80–90% confluency, they were detached and maintained in suspension on uncoated dishes, leading to Embryoid Body (EB) formation from day 6 onwards. EBs were selected and dissociated using a 1 ml pipette before being seeded onto Matrigel-coated dishes. The resulting NPCs were split approximately once per week using Accutase (Thermo Fisher Scientific, 07920) and expanded for at least 15 passages to obtain a pure NPC culture. The medium was refreshed every other day.

NPCs were cultured in expansion medium on Matrigel-coated dishes. For NPC differentiation into MNs, cells were cultured for 9 days in neuronal medium supplemented with 1 µM PMA. On day 2, 1 µM Retinoic acid (Stemcell Technologies, 72264) was added to the medium. The medium was

changed every other day. The culture medium was replaced every other day. After 9 days of differentiation, MNs were cryopreserved in CryoStor® CS10 (STEMCELL Technologies, 07930) and stored in liquid nitrogen for subsequent use in neuromuscular co-culture experiments.

## Neuromuscular co-cultures generation

Polydimethylsiloxane (PDMS) dishes were prepared following the protocol by Afshar Bakooshli et al. (2019) with minor modifications. Briefly, 3.5 cm Petri dishes (Sarstedt) were coated with liquid PDMS (Sigma, 761036-5EA), cured at 65°C, and embedded with dumbbell-shaped acrylic templates. After curing, Velcro anchors were affixed, and the dishes were sterilized using 70% ethanol and 15 min ultraviolet (UV) light before storage at room temperature. Prior to hydrogel seeding, PDMS molds were treated with 5% Pluronic acid (Thermo Fisher Scientific, P6866) at 4°C for 4 h to prevent cell adhesion.

For 3D neuromuscular co-cultures, 1.5 million skeletal myoblasts, 75,000 fibroblasts, and 500,000 MNs were suspended in a hydrogel composed of 20% Geltrex (Gibco, A1413302), 4 mg/ml fibrinogen, 0.8 U thrombin, and DMEM/F-12, in a final volume of 200 µl. This mixture was loaded into PDMS molds, fully covering the Velcro hooks, and incubated at 37°C for 5 min to solidify before adding medium. The 3D differentiation medium consisted of neuronal medium supplemented with 2% horse serum (Sigma, H1138), 10 ng/ml insulin (Merck, I9278), 2 mg/ml 6-aminocaproic acid (ACA) (Sigma, A2504), 2.5 µg/ml amphotericin B (Sigma, 15290-026), and 50 ng/ml agrin (R&D Systems, 550-AG-100). MNs derived from day 9 of the Reinhardt et al. protocol (iPSC line#1) were cultured in this medium with 10 ng/ml GDNF and BDNF, increased to 20 ng/ml after 2 days. MNs from day 6 of the Kroehne et al. protocol (iPSC line#2) received 3D differentiation medium supplemented with 200 µM ascorbic acid, 2 ng/ml GDNF, BDNF, and CNTF, 1 ng/ml TGFβ3, 200 µM dbcAMP, and 10 µM DAPT, with growth factors increased to 4 ng/ml after 2 days. Medium was changed every other day, and co-cultures were maintained until the designated experimental time points.

## RNA Isolation

Flash-frozen neuromuscular co-cultures were weighed, and 500 µl of QIAzol Lysis Reagent (Qiagen, lot no. 560012414) was added per 50 mg of tissue. The sample was transferred to a 2 ml Eppendorf tube, supplemented with a small amount of quartz sand, and homogenized using a Dounce homogenizer for 5 min on an ice bath or until no visible tissue fragments remained. Following homogenization, the sample was incubated at room temperature for 5 min and centrifuged at $3500 \times g$ for 15 min at 4°C. The supernatant was carefully transferred to a fresh Eppendorf tube, and 200 µl of chloroform was added. The mixture was vigorously shaken for 15 s, incubated at room temperature for 2–3 min, and then centrifuged at $12,000 \times g$ for 15 min at 4°C. The aqueous phase was transferred to a new tube, mixed thoroughly with 350 µl of 70% ethanol by vortexing, and subjected to further RNA purification using the NucleoSpin RNA kit (Macherey-Nagel, 740955.250). The extracted RNA samples were subsequently submitted to the core facility for RNA-seq.

## RNA sequencing & Data analysis

The concentration and the level of degradation (DV200) of the RNA were assessed using 5200 Fragment Analyzer (DNF-471-33 - SS Total RNA 15 nt, Agilent Technologies). Samples with DV200>30% were selected for further analysis. For cDNA preparation, the SMARTer® Stranded Total RNA-Seq Kit version 3 - Pico Input Mammalian (Takara) with UMIs was used with 5 ng of input DNase-treated RNA. The PCR protocol was optimized by using 12 PCR cycles. The quality of the dual-indexed libraries was checked using 5200 Fragment Analyzer (DNF-474-33-HS NGS Fragment 1-6000 bp, Agilent Technologies) and the average size was calculated at approximately 400 bp. The libraries were pooled at equimolar ratios and spiked with 1% PhiX control library. Sequencing was performed at 19-37 million read pairs/sample in paired-end mode with 54 and 68 nt read length for R1 and R2 reads, respectively, on the NextSeq 2000 platform (Illumina) using a P2 (100 cycles) sequencing kit. Demultiplexed FASTQ files were generated with bcl-convert version 4.3.6 (Illumina).

Unique molecular identifier (UMI) sequences were extracted from the beginning of R2 reads using UMI-tools (Smith et al., 2017) version 1.1.1

(parameters: extract –extract-method=regex –bc-pattern2=^(?P<umi_1>.{8})(?P<discard_1>.{6}).* for all read pairs). The resulting read pairs were quality- and adapter-trimmed via Cutadapt (Martin, 2011) version 2.5 in paired-end mode (parameters: –nextseq-trim=20 -m 1 -a NNNNNNNNNNNNNNNAGATCGGAAGAGCACACGTCTGAACTCC-AGTCAC;min_overlap=17 -A AGATCGGAAGAGCGTCGTGTAGGG-AAAGAGTGT). Processed read pairs were aligned to the human genome (GCF_000001405.40/GRCh38.p14, primary assembly and mitochondrion) using STAR (Dobin et al., 2013) version 2.7.2b with default parameters but including transcript annotations from RefSeq annotation version RS_2023_03 for GRCh38.p14. Afterwards, aligned read pairs were deduplicated with UMI-tools version 1.1.1 (parameters: dedup –paired –random-seed 123456789 –multimapping-detection-method=NH). The resulting alignment files were subsequently used for gene expression quantification via featureCounts version 1.6.4 from the Subread package (Liao et al., 2014). Only fragments with both ends aligned to the same chromosome and strand were quantified on exon level and summarized to a fragment count for each gene. For this, multi-mapping and multi-overlapping fragments were counted strand-specific and reversely stranded with a fractional count for each aligned fragment and overlapping feature (parameters:-p -B -C -s 2 -t exon -M -O –fraction). The count output was used to identify differentially expressed genes using DESeq2 (Love et al., 2014) version 1.24.0. Read counts were normalized by DESeq2 and fold-change shrinkage was applied by setting the parameter "betaPrior=TRUE". Differential expression of genes was assumed at an adjusted $P$-value (padj) after Benjamini-Hochberg correction <0.05 and |log2FoldChange| ≥1. For GO term analysis, we used the Database for Annotation, Visualization and Integrated Discovery (DAVID) (Huang da et al., 2009). We used the 42.059 transcripts expressed with an average normalized read count >0.5 as reference datasets.

## Acknowledgements

We thank Janina Dix and Alexander Krybus for excellent technical support. We thank Panagiota Arampatzi for support with RNA-seq. We thank the Core Unit Systems Medicine for performing the RNA-seq experiments.

## Competing interests

The authors declare no competing or financial interests.

## Author contributions

Conceptualization: P.L.; Formal analysis: P.L., N.J.G., T.B., M.B.; Funding acquisition: P.L.; Methodology: N.J.G., B.H., M.S.; Resources: P.L., B.S.; Supervision: P.L., B.S.; Validation: P.L., N.J.G.; Visualization: P.L.; Writing – original draft: P.L., N.J.G.; Writing – review & editing: P.L., N.J.G., B.H.

## Funding

P.L. was supported by the BMBF project VORAN-2, 16LW066, and by funding of the SET foundation. This work was supported by the IZKF at the University Hospital of Würzburg (project Z-6). Open Access funding was provided by the University of Würzburg in the funding programme Open Access Publishing. Deposited in PMC for immediate release.

## Data and resource availability

The RNA-seq data have been deposited in NCBI's Gene Expression Omnibus (GEO) with dataset identifier GSE297790.

## First person

This article has an associated First Person interview with the co-first authors of the paper.

## Peer review history

The peer review history is available online at https://journals.biologists.com/bio/lookup/doi/10.1242/bio.062196.reviewer-comments.pdf

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
