## [Peer Review File · Biology Open]

Temporal transcriptomic profiling of human 3D neuromuscular co-cultures

Neha Jadhav Giridhar, Bitá Hambrecht, Maren Schenke, Bettina Seeger, Thorsten Bischler, Michael Briese and Patrick Lüningschrör
DOI: 10.1242/bio.062196

Editor: Tristan Rodríguez

Review timeline

Original submission:	29 May 2025
Editorial decision:	27 June 2025
First revision received:	4 August 2025
Accepted:	6 August 2025

Original submission

First decision letter

MS ID#: bio.062196

MS TITLE: Temporal transcriptomic profiling of human 3D neuromuscular co-cultures

AUTHORS: Patrick Lüningschrör; Neha Jadhav Giridhar; Bitá Hambrecht; Maren Schenke; Bettina Seeger; Thorsten Bischler; Michael Briese

I have now received all the referees' reports on the above manuscript, and have reached a decision. I am sorry to say that the outcome is not a positive one. The referees' comments are appended below, or you can access them online: please go to .

As you will see, the referees raise some significant concerns about your paper, and are not strongly in favour of publication. The primary concern is that your study presents a descriptive transcriptomic analysis without the functional validation necessary to demonstrate that the observed molecular changes reflect genuine neuromuscular junction maturation. The reliance on bulk RNA sequencing, and lack of functional readouts such as electrophysiology, immunostaining, or contractility assays, limits the interpretability and impact of the findings. The referees agree that in its current form the work does not sufficiently advance our understanding to be of broad interest to the neuromuscular research community. Having looked at the manuscript myself, I am afraid I agree with their views, and I must therefore, reject your paper.

Reviewer 1

Comments to Author

The manuscript by Giridhar and colleagues presents in vitro developmental transcriptomic analyses on 3-dimensional muscle/motor neuron co-cultures using a previously reported experimental setup (Massih et al., 2023). They perform bulk RNA sequencing at three timepoints—1 week, 3 weeks, and 6 weeks—and identify gene sets with distinct patterns of expression changes over time. Characterization of these sets reveals differentiation of progenitors into terminally differentiated cell types, production of extracellular matrix (ECM) components, and establishment of synapses.

Finally, the authors attempt to demonstrate the robustness of their findings by conducting transcriptomic analysis on 3D co-cultures using neurons derived from an alternative iPSC source.

There is no doubt about the merits of evaluating in vitro myotube/neuron co-cultures; it is critical to understand if these systems recapitulate in vivo physiology (and to what extent), if they exhibit species specific differences in NMJ development and structure, and to identify the dynamics of this developmental process. In this regard, the authors have already developed excellent resources in Massih et al. (2023), identifying periods during which neurite outgrowth and muscle fiber contractions are dynamic. Thus, they chose these three previously characterized timepoints for transcriptomic analysis. However, the range of analyses performed in this manuscript fails to sufficiently answer the above questions: as the dataset is from a bulk source, there is a general lack of depth on a variety of concerns, including 1) which cell types express which genes, 2) cross-validation of identified differential genes using alternate approaches, 3) comparison to in vivo changes, 4) whether these changes truly reflect bona fide NMJ development, and 5) comparison with phenotyping of the co-cultures. This lack of depth is also evident in the replication of findings using an alternative iPSC source (Figure 4), for which only summary-level analyses have been included. Overall,

Major concerns

1. The general analyses presented here seem somewhat shallow. For example, as the co-culture begins with myoblasts, fibroblasts, and NPCs, it would be prudent to distinguish gene expression changes related to terminal differentiation into muscles/motor neurons from those happening during maturation of differentiated muscles/neurons over time. Similarly, it would be useful to further characterize the gene ontologies based on changes expected to be related to differentiation versus maturation. An additional timepoint at DIV0 would facilitate these analyses.
2. Splicing changes over time in vitro should also be reported and characterized.
3. This manuscript relies on a single readout of gene expression—bulk RNA sequencing. There are no independent confirmations and limited understanding of which cell types in the co-culture are driving gene expression changes. Cross-validation of gene expression changes using FISH/RNA scope and immunostaining would strengthen the findings and provide greater resolution about cell-type-specific gene expression changes.
4. The introduction of this manuscript orients the reader to shared and human-specific features of NMJs. However, gene expression profiling was not used to identify commonalities and differences with similar datasets from other species (such as rodents, where published muscle/motor neuron data are available). Such analyses, which need not require the generation of additional datasets, would demonstrate species-specific and shared changes, and would help champion the value of the reported datasets.
5. While the authors note thousands of gene expression changes, including many changes in expression of AChR, it is not clear if changes in transcripts reflect maturation of synaptic complexes at the NMJ level. Immunostaining, morphometric, pharmacological, or functional evaluation of NMJs over time would strengthen the hypothesis that the transcriptomic data reflect maturation of NMJs over time. Similarly, comparison with neuron or muscle monocultures would strengthen the idea that the identified gene expression changes are truly related to NMJ formation in the co-culture system.
6. Characterization of the co-culture system in Massih et al. (2023) showed that contractions and neurite outgrowth plateaued or decreased after 3 weeks in culture. This seems directly in contradiction with findings here, where synaptic maturation occurs between 3 and 6 weeks. Wouldn't these changes also alter muscle contractions?

Minor concerns

7. The labels for $\uparrow\downarrow$ and $\downarrow\uparrow$ in Figure 1F seem incorrect.

8. What are the authors trying to convey in the following sentence? It is not clear: "The significant heterogeneity in NMJ morphology raises questions about how the form and function of NMJs in one species can be applied to another species."

Reviewer 2

Comments to Author

Giridhar et al report the generation of a 3-D human neuromuscular co-culture of hiPSC-derived motor neurons with primary human myoblasts. Using longitudinal bulk transcriptomics, the authors trace a gene-expression trajectory that recapitulates canonical milestones of neuromuscular-junction (NMJ) development, beginning with myoblast fusion and presynaptic specification and progressing to cholinergic signaling and postsynaptic specialization. They propose the platform as a human-specific model for studying NMJ formation and neuromuscular disease.

Although the bulk RNA seq data provides a useful temporal reference map, the model's advancement over previously published hiPSC-muscle co-cultures is not demonstrated. Similar systems, many of which use fully isogenic, self-organized neuromuscular models, already document molecular and functional maturation of human NMJs. In this study, the evidence is limited to transcriptional profiles; there are no electrophysiological recordings, synaptic transmission assays, contractility measurements, or pharmacological perturbations to confirm functional NMJs. Without such validation, it remains uncertain whether the co-culture can faithfully model NMJ physiology or serve as a robust platform for disease modeling and drug testing. Incorporating functional readouts and clarifying the rationale for mixing primary muscle with hiPSC-derived neurons would markedly strengthen the study's novelty and translational relevance.

Major Concerns

Although the study offers a temporal transcriptomic analysis of a 3D neuromuscular co-culture system, the central model, which is co-culturing hiPSC-derived motor neurons with skeletal muscle derived from primary human myoblasts, raises concerns about physiological relevance. The use of primary muscle, rather than iPSC-derived muscle from the same genetic background, introduces a non-isogenic and developmentally asynchronous component that may compromise the relevance of neuromuscular interactions. This mismatch limits the system's utility for patient-specific disease modeling and may obscure critical developmental processes that rely on temporally coordinated signaling between motor neurons and muscle. Moreover, self-organized neuromuscular models that integrate both lineages from a common iPSC source have already been described in the literature, offering more developmentally relevant and scalable systems. The manuscript does not clearly explain the rationale or advantage of using primary muscle in this co-culture context, and there are no clear functional benefits over the existing, fully stem cell-derived platforms.

There is a lack of imaging data and functional validation to characterize the model. Although the authors have included detailed transcriptomic data, the manuscript lacks direct evidence of functional NMJ formation. Key functional readouts, such as α -bungarotoxin labeling of NMJs, electrophysiology, and muscle contraction assays, are absent. This omission weakens the central claim that the system robustly recapitulates NMJ maturation and limits the translational value of the model.

The use of non-isogenic primary human myoblasts (instead of iPSC-derived muscle) introduces heterogeneity and significantly limits the applicability of the system to disease modeling, especially in genetic disorders affecting both neuronal and muscular compartments. The platform, therefore, may not support high-throughput or personalized approaches as effectively as fully iPSC-based systems.

The study uses bulk RNA sequencing, which fails to resolve cell-type-specific gene expression dynamics. The field is increasingly moving toward single-nucleus or single-cell RNA-seq, especially for complex co-culture systems. This limitation reduces the ability to precisely attribute molecular events to motor neurons versus muscle fibers, or to dissect cell-cell interaction mechanisms.

Despite suggesting that the platform is suitable for disease studies, the manuscript does not include any disease modeling. The absence of such experiments undermines the proposed use of the model in translational research.

Reviewer 3

SUMMARY OF THE ADVANCE MADE IN THIS PAPER AND ITS POTENTIAL SIGNIFICANCE TO THE FIELD

This paper presents sequencing data on a model of human neuromuscular junction in vitro and proposes that its use could represent a valuable resource within the field. The author use an already published in vitro system for neuromuscular junction formation, already used for 2 prior publications in 2019 and 2023 and previously fully characterised, and extract RNA for sequencing at 3 developmentally relevant timepoints. The results are used to infer developmental match with the in vivo counterpart.

SUGGESTIONS TO AUTHORS

The field of neuromuscular disease and tools for neuromuscular modelling in vitro, especially in humanised models, is an important one, and advancement of technology platform in this area can have sizable impact, as they would allow better modelling of neuromuscular disorders. This paper presents effectively one experiment with bulk RNA seq at 3 different timepoints, on a model previously published and characterised already, which focused on the expression of different transcript that according to the authors indicate specific patterns of maturation for human NMJs. While it is indeed true that human NMJs and dataset on their maturation and transcriptional (as well as functional profile) are not a widely available resource and need more development to be established, this reviewer is not convinced that the study in the present form constitute a significantly valuable and established (or useful) resource for this field, at least in its present form.

This entire study is one experiment, effectively, and does in no way advance or improve the published NMJ model itself. The data presented is entirely descriptive, which in itself wouldn't be a problem, but the author have provided no comparison with any other established model, in vivo direct systems, established murine NMJs in vivo or in vitro, or any other relevant comparison. Indeed, some of the observed pattern of expression suggest a potential coherence between the model and the in vivo hNMJ development (albeit, in the most vague and general of terms), but to prove that this system can be used as a valuable resource there need to be more validation.

The paper is also not helped by the choice of writing, which completely forgoes any explanation of the culture system, merely cites it, and presents Results and Discussion joined, which results in virtually no discussion at all on the potential pros and cons, considerations etc. on this system.

In its current form, this feels like either an additional set of figures for the characterisation of the model, or the beginning of a study, but it is not a complete and useful resource at this stage, in my opinion.

Moreover, the authors only use 2 ips lines for validation, and while the trends observed are similar there are clearly variations that would suggest at least 3-4 lines (as it is usually done in this studies) would have been ideal.

First revision

Author response to reviewers' comments

We are grateful for the reviewers' time and effort in evaluating our manuscript. The insightful comments and suggestions have greatly helped us improve the clarity and rigor of our work. Below, we provide point-by-point responses to each of the reviewers' concerns. Our responses are in **red font**. Page and line numbers point to changes within the revised manuscript. In the manuscript, revised text paragraphs are highlighted in **green**.

Reviewer 1:

The manuscript by Giridhar and colleagues presents *in vitro* developmental transcriptomic analyses on 3-dimensional muscle/motor neuron co-cultures using a previously reported experimental setup (Massih et al., 2023). They perform bulk RNA sequencing at three timepoints—1 week, 3 weeks, and 6 weeks—and identify gene sets with distinct patterns of expression changes over time. Characterization of these sets reveals differentiation of progenitors into terminally differentiated cell types, production of extracellular matrix (ECM) components, and establishment of synapses. Finally, the authors attempt to demonstrate the robustness of their findings by conducting transcriptomic analysis on 3D co-cultures using neurons derived from an alternative iPSC source.

There is no doubt about the merits of evaluating *in vitro* myotube/neuron co-cultures; it is critical to understand if these systems recapitulate *in vivo* physiology (and to what extent), if they exhibit species specific differences in NMJ development and structure, and to identify the dynamics of this developmental process. In this regard, the authors have already developed excellent resources in Massih et al. (2023), identifying periods during which neurite outgrowth and muscle fiber contractions are dynamic. Thus, they chose these three previously characterized timepoints for transcriptomic analysis. However, the range of analyses performed in this manuscript fails to sufficiently answer the above questions: as the dataset is from a bulk source, there is a general lack of depth on a variety of concerns, including 1) which cell types express which genes, 2) cross-validation of identified differential genes using alternate approaches, 3) comparison to *in vivo* changes, 4) whether these changes truly reflect bona fide NMJ development, and 5) comparison with phenotyping of the co-cultures. This lack of depth is also evident in the replication of findings using an alternative iPSC source (Figure 4), for which only summary-level analyses have been included. Overall,

Major concerns

1. The general analyses presented here seem somewhat shallow. For example, as the co-culture begins with myoblasts, fibroblasts, and NPCs, it would be prudent to distinguish gene expression changes related to terminal differentiation into muscles/motor neurons from those happening during maturation of differentiated muscles/neurons over time. Similarly, it would be useful to further characterise the gene ontologies based on changes expected to be related to differentiation versus maturation. An additional timepoint at DIV0 would facilitate these analyses.

We thank the reviewer for highlighting the importance of distinguishing early lineage commitment from later maturation events. Unfortunately, a DIV0 sample was not collected in the original experimental design and thus cannot be retrospectively added. We discussed this limitation in the newly added discussion section. Furthermore, we addressed this point by reorganizing our GO-term enrichment results into three biologically meaningful phases— Early myogenesis, Synapse assembly, and Functional maturation—and added the following paragraph to the Results (p.5, lines 16-31). We have also added a brief statement in the Discussion (p.8, lines 32-37) acknowledging the lack of a DIV0 sample as a limitation and highlighting this phase-based framework as our chosen strategy to interpret temporal gene-expression dynamics.

“Overall, we classified the most enriched biological processes into three principal phases that mirror the established *in vivo* sequence of hNMJ development. During the early myogenesis phase, the enrichment of terms such as cell division, mitotic cell cycle, chromosome segregation, mitotic spindle assembly, mitotic cytokinesis, extracellular matrix (ECM) organization, signal transduction, cAMP-mediated signaling underscores myoblast proliferation, ECM remodeling, and readiness for fusion. The synapse assembly phase encompasses GO terms including synapse assembly, chemical synaptic transmission, acetylcholine receptor signaling, cholinergic synaptic transmission, regulation of postsynaptic membrane potential, cell-cell adhesion, homophilic cell adhesion, GPCR-coupled glutamate receptor signaling, cell-cell signaling, and cell adhesion, highlighting the formation of both pre- and post-synaptic structures and adhesion machinery. In the functional maturation phase, enriched processes such as sodium/potassium/chloride ion transport, Ca²⁺-dependent cell-cell adhesion, regulation of membrane potential, monoatomic ion membrane transport, lipid metabolic process, sarcomere organization/striated muscle contraction/muscle

contraction reflect the development of electrical excitability, ion handling, and contractile capability. These three phases - early myogenesis, synapse assembly, and functional maturation - thereby offer a streamlined, stage-focused framework for interpreting the time-dependent shifts in our 3D neuromuscular co-cultures.”

“The absence of an early reference point (day *in vitro* 0, DIV0) is another limitation of this study, which restricts our ability to discriminate between transcriptional programs associated with initial differentiation versus those tied to later maturation. As a result, early myogenic or synaptogenic events occurring before our 1-week baseline may be underrepresented. To partially mitigate this limitation, we reorganized GO-term enrichment results into three biologically relevant stages – early myogenesis, synapse assembly, and functional maturation.”

2. Splicing changes over time *in vitro* should also be reported and characterized.

We thank the reviewer for highlighting the importance of alternative splicing. Although isoform-level insights would certainly enrich our understanding, our existing RNA-seq libraries were generated using total-RNA protocols and sequencing parameters optimized for gene-level differential expression. Robust analysis of splicing events typically requires poly(A)-enriched or long-read data and specialized pipelines that were beyond the scope of this study. We have therefore noted in the Discussion (p.9, lines 9-12) that:

“In parallel, future work will also address transcript-level complexity, as alternative splicing represents an important dimension of transcriptome regulation not examined here due to library preparation and sequencing constraints.”

3. This manuscript relies on a single readout of gene expression—bulk RNA sequencing. There are no independent confirmations and limited understanding of which cell types in the co-culture are driving gene expression changes. Cross-validation of gene expression changes using FISH/RNA scope and immunostaining would strengthen the findings and provide greater resolution about cell-type-specific gene expression changes.

We agree with the reviewer that a validation of the transcriptional changes by FISH/RNA scope or immunostaining would strengthen our conclusion and allow determining cell-type-specific gene expression changes. However, we cannot provide these data in the current study. Therefore, we discussed this limitation in the discussion (p.8, lines 23-30).

“While our bulk RNA-seq data delineate coordinated waves of myogenic, neuronal, and synaptic gene expression, this approach inherently averages signals across heterogeneous cell populations, thereby obscuring lineage-specific contributions. Because bulk RNA-seq does not resolve cell-type-specific expression, it precludes attribution of transcriptional changes to defined lineages. Future studies employing single-nucleus RNAseq (snRNA-seq) could address this limitation and offer improved cellular resolution. Alternatively, RNA-FISH or immunostaining approaches could be used to provide spatial and cell-type-specific validation. However, such experiments were beyond the scope of the present study and represent valuable opportunities for future work.”

4. The introduction of this manuscript orients the reader to shared and human-specific features of NMJs. However, gene expression profiling was not used to identify commonalities and differences with similar datasets from other species (such as rodents, where published muscle/motor neuron data are available). Such analyses, which need not require the generation of additional datasets, would demonstrate species-specific and shared changes, and would help champion the value of the reported datasets.

We thank the reviewer for this suggestion. However, our study did not aim for a species comparison. We apologize if the introduction orients the reader in this direction. Therefore, we shortened this paragraph and extended the part regarding human neuromuscular cell culture systems. We have streamlined the introduction to succinctly note well-established human-rodent NMJ differences and clarify our focus on developing a human-specific model. The revised text now reads (p.2, lines 4-10):

“Human NMJs (hNMJs) exhibit distinctive morphological and molecular features—notably smaller

presynaptic terminals and quantal vesicle size relative to muscle fiber diameter, combined with deeper postsynaptic infoldings that enlarge synaptic surface area (Slater, 2017). In addition, their synaptic proteome composition is distinct, with divergent localization of active zone proteins such as SNAP-25 (Jones et al., 2017). These morphological and proteomic differences from commonly used rodent models underscore the necessity for dedicated hNMJ systems.”

In addition, we now explicitly acknowledge in the discussion (p.9, lines 3-6) that while our study provides key insights into human NMJ maturation, no direct comparative analysis with rodent datasets was performed, and future comparative studies would help contextualize species-specific features and benchmark *in vitro* human models against established rodent paradigms.

“Our current focus on a human-specific co-culture model offers unique insights into hNMJ development but lacks a direct comparative analysis with rodent datasets. Such comparisons could contextualize species-specific features and serve as benchmarks for evaluating the fidelity of *in vitro* human systems.”

5. While the authors note thousands of gene expression changes, including many changes in expression of AChR, it is not clear if changes in transcripts reflect maturation of synaptic complexes at the NMJ level. Immunostaining, morphometric, pharmacological, or functional evaluation of NMJs over time would strengthen the hypothesis that the transcriptomic data reflect maturation of NMJs over time. Similarly, comparison with neuron or muscle monocultures would strengthen the idea that the identified gene expression changes are truly related to NMJ formation in the co-culture system.

We appreciate the reviewer’s emphasis on functional validation. In the newly added discussion section, we now acknowledge the importance of functional assays. Our previous reports in similar human neuromuscular co-culture systems (Afshar Bakooshi et al., 2019; Massih et al., 2023) have demonstrated that transcriptional upregulation of synaptic components such as AChR subunits, Rapsyn, and MuSK correlates with the emergence of functional NMJs. This supports the interpretation that the transcriptomic dynamics observed here likely reflect bona fide early-stage NMJ formation processes. We also acknowledge that the absence of orthogonal validation in our system using RNA-FISH, RNAscope, or immunostaining remains a limitation. Additionally, we also acknowledge that future studies comparing our co-culture system with neuron-only or muscle-only monocultures would further strengthen the inference that the observed transcriptional programs are NMJ-specific. These clarifications and additions appear in the revised discussion (p.8, lines 23-30; p.8-9, lines 38-2).

“While our bulk RNA-seq data delineate coordinated waves of myogenic, neuronal, and synaptic gene expression, this approach inherently averages signals across heterogeneous cell populations, thereby obscuring lineage-specific contributions. Because bulk RNA-seq does not resolve cell-type-specific expression, it precludes attribution of transcriptional changes to defined lineages. Future studies employing single-nucleus RNA-seq (snRNA-seq) could address this limitation and offer improved cellular resolution. Alternatively, RNA-FISH or immunostaining approaches could be used to provide spatial and cell-type-specific validation. However, such experiments were beyond the scope of the present study and represent valuable opportunities for future work.”

“However, imaging-based approaches combined with functional assays, such as electrophysiological recordings or contractility measurements, will be essential for validating these molecular observations.”

6. Characterization of the co-culture system in Massih et al. (2023) showed that contractions and neurite outgrowth plateaued or decreased after 3 weeks in culture. This seems directly in contradiction with findings here, where synaptic maturation occurs between 3 and 6 weeks. Wouldn’t these changes also alter muscle contractions?

We appreciate the reviewer’s careful reading and the opportunity to clarify this point. In Massih et al. 2023, functional characterization of healthy (wild-type) 3D neuromuscular co-cultures showed that axon growth peaked around day 21 and then stabilized through day 42. Glutamate-induced muscle contractions similarly peaked at ~3 weeks and exhibited a modest

decline by 6 weeks. These results indicate that functional NMJ connectivity was largely preserved over this later period in healthy co-cultures. Our observation that transcriptomic maturation continues between weeks 3 and 6 is therefore not necessarily inconsistent with Massih et al. (2023). The late-phase transcriptomic changes we report may reflect qualitative synaptic remodeling (e.g., postsynaptic scaffold reorganization or acetylcholine receptor subunit switching) rather than further increases in bulk contractile output. We agree that future work combining longitudinal functional assays with molecular profiling will be valuable to more precisely resolve the relationship between synaptic maturation and functional performance.

Minor concerns

7. The labels for $\uparrow\downarrow$ and $\downarrow\uparrow$ in Figure 1F seem incorrect.

We thank the reviewer for catching this error. We have corrected the $\uparrow\downarrow$ and $\downarrow\uparrow$ labels in Figure 1F as indicated.

8. What are the authors trying to convey in the following sentence? It is not clear: "The significant heterogeneity in NMJ morphology raises questions about how the form and function of NMJs in one species can be applied to another species."

We changed the sentence in introduction (p.2, lines 3-4) as follows:

"Although the principal organization of vertebrate/mammalian NMJs shares essential features, there are considerable interspecies differences in the NMJ morphology."

Reviewer 2

Giridhar et al report the generation of a 3-D human neuromuscular co-culture of hiPSC- derived motor neurons with primary human myoblasts. Using longitudinal bulk transcriptomics, the authors trace a gene-expression trajectory that recapitulates canonical milestones of neuromuscular-junction (NMJ) development, beginning with myoblast fusion and presynaptic specification and progressing to cholinergic signaling and postsynaptic specialization. They propose the platform as a human-specific model for studying NMJ formation and neuromuscular disease.

Although the bulk RNA seq data provides a useful temporal reference map, the model's advancement over previously published hiPSC-muscle co-cultures is not demonstrated. Similar systems, many of which use fully isogenic, self-organized neuromuscular models, already document molecular and functional maturation of human NMJs. In this study, the evidence is limited to transcriptional profiles; there are no electrophysiological recordings, synaptic transmission assays, contractility measurements, or pharmacological perturbations to confirm functional NMJs. Without such validation, it remains uncertain whether the co- culture can faithfully model NMJ physiology or serve as a robust platform for disease modeling and drug testing. Incorporating functional readouts and clarifying the rationale for mixing primary muscle with hiPSC-derived neurons would markedly strengthen the study's novelty and translational relevance.

Major Concern

Although the study offers a temporal transcriptomic analysis of a 3D neuromuscular co-culture system, the central model, which is co-culturing hiPSC-derived motor neurons with skeletal muscle derived from primary human myoblasts, raises concerns about physiological relevance. The use of primary muscle, rather than iPSC-derived muscle from the same genetic background, introduces a non-isogenic and developmentally asynchronous component that may compromise the relevance of neuromuscular interactions. This mismatch limits the system's utility for patient-specific disease modeling and may obscure critical developmental processes that rely on temporally coordinated signaling between motor neurons and muscle. Moreover, self-organized neuromuscular models that integrate both lineages from a common iPSC source have already been described in the literature, offering more developmentally relevant and scalable systems. The manuscript does not clearly explain the rationale or advantage of using primary muscle in this co-culture context, and there are no clear functional benefits over the existing,

fully stem cell-derived platforms.

We appreciate the reviewer's insightful comment regarding the choice of primary human myoblast-derived muscle rather than isogenic iPSC-derived muscle. Our decision was guided by practical and biological considerations relevant to the aims of this study. Primary myoblasts consistently and robustly differentiate into aligned, contractile myotubes within two weeks, exhibiting reproducible morphology and contractile behavior across donors (Afshar Bakooshli et al., 2019; Massih et al., 2023). In contrast, iPSC-derived myotubes often require extended culture periods (>4-6 weeks), display variable fusion efficiency, and exhibit inconsistent contractile properties (Borchin et al., 2013; Pinton et al., 2023), which can complicate longitudinal transcriptomic profiling.

Because our goal was to generate a high-temporal-resolution transcriptomic map of human NMJ development under stable and reproducible conditions, primary myoblast-based muscle offered a technically reliable platform that reduced biological variability unrelated to NMJ formation. We acknowledge, however, that this approach introduces developmental asynchrony and lacks the genetic matching of patient-derived, fully isogenic systems, which limits its application for certain disease-modeling contexts.

We agree that isogenic iPSC-derived muscle-MN co-cultures, or fully self-organized neuromuscular organoids, offer important advantages for modeling genetically driven neuromuscular diseases and recapitulating temporally coordinated developmental programs. Incorporating such systems will be a priority for future work, particularly as differentiation protocols improve in reproducibility and yield. Future studies can compare transcriptomic trajectories between primary myoblast-based co-cultures and fully isogenic iPSC-derived systems to directly assess the impact of lineage matching on NMJ development and disease modeling fidelity.

We have now compared the advantages and disadvantages of various models in our introduction (p2. lines 11-28) and discussion (p.9 lines 16-23):

“Over the past decade, the neuromuscular modeling field has evolved from 2D MN-myotube assays to complex three-dimensional constructs that more closely mimic native tissue architecture. Early 2D co-cultures provided valuable insights into synapse formation. They were used to model conditions like Duchenne muscular dystrophy (Maffioletti et al., 2018), myasthenia gravis (Steinbeck et al., 2016), and amyotrophic lateral sclerosis (ALS) (Stoklund Dittlau et al., 2021; Bademosi et al., 2023; Guo et al., 2020), but they lack the mechanical cues and cell-matrix interactions of muscle *in vivo*. 3D co-cultures address these limitations by embedding iPSC-derived MNs and primary human myoblast-derived fibers in a supportive matrix, enabling the formation of aligned myofibers, spontaneous contraction, and functional NMJs (Afshar Bakooshli et al., 2019; Massih et al., 2023).

Fully isogenic organoid models grown entirely from iPSCs offer the added benefit of synchronized development and inclusion of accessory cells such as SCs and endothelial populations (Pereira et al., 2021; Faustino Martins et al., 2020), but they can suffer from variability in size, necrotic centers, and lengthy, complex differentiation protocols (Leng et al., 2023; Yang et al., 2025). Fused “assembloids” combine separate muscle and neural organoids to enhance synaptic connectivity and coordinated contractions, yet they remain challenging to scale and reproduce consistently (Yang et al., 2025). More recently, biohybrid spheroids—incorporating conductive materials and microvascular networks have demonstrated real-time electrophysiological readouts in ALS disease modeling (Shin et al., 2025).

“The use of primary human myoblasts offers a reliable platform, given their consistent differentiation into contractile myotubes within two weeks (Afshar Bakooshli et al., 2019; Massih et al., 2023). This contrasts with iPSC-derived myotubes, which often require extended culture durations and demonstrate variable fusion efficiency (Borchin et al., 2013; Pinton et al., 2023). While the co-culture system used here provides a simplified yet reproducible model, more complex approaches – such as neuromuscular organoids, assembloids, or biohybrid constructs – introduce greater cellular diversity and enable embedded biosensing but often encounter challenges related to variable maturation, reduced viability, and technical complexity (Leng et al., 2023; Yang et al., 2025; Shin et al., 2025).”

There is a lack of imaging data and functional validation to characterize the model. Although the

authors have included detailed transcriptomic data, the manuscript lacks direct evidence of functional NMJ formation. Key functional readouts, such as α -bungarotoxin labeling of NMJs, electrophysiology, and muscle contraction assays, are absent. This omission weakens the central claim that the system robustly recapitulates

We agree that functional assays are critical for confirming NMJ formation. While such assays were not performed in the present study, prior reports using nearly identical co-culture conditions have demonstrated spontaneous contractions and α -bungarotoxin-positive NMJs (Afshar Bakooshli et al., 2019; Massih et al., 2023). We now cite these works prominently in the Introduction (p.2, lines 34-38) and discussion (p.8, lines 27-31) to provide context, and we explicitly note that functional validation remains an essential direction for future work.

“While functional assays were not performed here, prior studies using nearly identical co-culture conditions have demonstrated spontaneous contractions and α -bungarotoxin-positive NMJs (Afshar Bakooshli et al., 2019; Massih et al., 2023).”

“Although functional assays were not performed in this study, prior reports using nearly identical co-culture conditions have demonstrated spontaneous contractions and α -bungarotoxin-positive NMJs (Afshar Bakooshli et al., 2019; Massih et al., 2023), supporting the expectation that our system can achieve functional connectivity.”

The use of non-isogenic primary human myoblasts (instead of iPSC-derived muscle) introduces heterogeneity and significantly limits the applicability of the system to disease modeling, especially in genetic disorders affecting both neuronal and muscular compartments. The platform, therefore, may not support high-throughput or personalized approaches as effectively as fully iPSC-based systems.

We appreciate the reviewer’s insightful comment. We now explicitly acknowledge in the revised manuscript (p.9-10, lines 37-1) that the use of non-isogenic myoblasts **may limit** the system’s suitability for modeling neuromuscular diseases with defined genetic backgrounds, particularly those affecting both muscle and motor neurons. Our choice to use primary myoblasts was driven by their reproducible and efficient differentiation into contractile myotubes, as supported by prior studies. Nevertheless, we agree that future iterations incorporating isogenic iPSC-derived muscle and neurons will further enhance the system’s applicability for patient-specific and high-throughput disease modeling.

“Finally, future adaptations incorporating isogenic, patient-derived iPSC MNs and muscle cells will facilitate disease modeling, especially in contexts where developmental timing and lineage-matching are critical for accurate phenotypic manifestation.”

The study uses bulk RNA sequencing, which fails to resolve cell-type-specific gene expression dynamics. The field is increasingly moving toward single-nucleus or single-cell RNA-seq, especially for complex co-culture systems. This limitation reduces the ability to precisely attribute molecular events to motor neurons versus muscle fibers, or to dissect cell-cell interaction mechanisms.

We agree with the reviewer that bulk RNA-seq lacks cell-type resolution and now explicitly acknowledge this limitation in the manuscript (p.8, lines 23-30; p.9, lines 6-9). In line with the reviewer’s suggestion, we note that future studies incorporating single-nucleus RNA-seq, as well as spatial validation techniques such as RNA-FISH or immunostaining, will enable more precise dissection of lineage-specific transcriptional dynamics and intercellular signaling. We appreciate this valuable perspective and have highlighted it as a key direction for future work.

“While our bulk RNA-seq data delineate coordinated waves of myogenic, neuronal, and synaptic gene expression, this approach inherently averages signals across heterogeneous cell populations, thereby obscuring lineage-specific contributions. Because bulk RNA-seq does not resolve cell-type-specific expression, it precludes attribution of transcriptional changes to defined lineages. Future studies employing single-nucleus RNA sequencing (snRNA-seq) could address this limitation and offer improved cellular resolution. Alternatively, RNA-FISH or immunostaining approaches could be used to provide spatial and cell-type-specific validation. However, such experiments were beyond the scope of the present study and represent valuable opportunities for future work.”

“Implementing snRNA-seq or scRNA-seq across embryonic, neonatal, and adult tissues – including cell types such as PAX7⁺ satellite cells, HES1⁺/TRIB1⁺ progenitors, endothelial cells, and immune populations – could further delineate developmental stage-specific contributions and enhance our understanding of hNMJ maturation (Cai et al., 2023; Xu et al., 2023). “

Despite suggesting that the platform is suitable for disease studies, the manuscript does not include any disease modeling. The absence of such experiments undermines the proposed use of the model in translational research.

We appreciate the reviewer’s comment and have adjusted the manuscript’s framing to clarify that the current study does not include disease modeling experiments. Instead, our primary aim is to generate a temporal transcriptomic reference for human NMJ development, which can serve as a foundation for future disease-focused investigations. We now explicitly state this in the Introduction (p.3, lines 4-6) and Discussion (p.10, lines 1-4). For example:

“Together, these data demonstrate that human 3D neuromuscular co-cultures recapitulate transcriptional features of embryonic NMJ development and provide a temporal reference framework to inform future studies of NMJ formation and disease.”

“By addressing current limitations through multi-omic integration, spatial validation, functional assays, and advanced biomechanical engineering—and by benchmarking against in vivo and rodent models—this platform can be utilized for studying human NMJ development, disease, and therapeutic discovery.”

Reviewer 3

SUMMARY OF THE ADVANCE MADE IN THIS PAPER AND ITS POTENTIAL SIGNIFICANCE TO THE FIELD

This paper presents sequencing data on a model of human neuromuscular junction in vitro and proposes that its use could represent a valuable resource within the field. The author use an already published in vitro system for neuromuscular junction formation, already used for 2 prior publications in 2019 and 2023 and previously fully characterised, and extract RNA for sequencing at 3 developmentally relevant timepoints. The results are used to infer developmental match with the in vivo counterpart.

SUGGESTIONS TO AUTHORS

The field of neuromuscular disease and tools for neuromuscular modelling in vitro, especially in humanised models, is an important one, and advancement of technology platform in this area can have sizable impact, as they would allow better modelling of neuromuscular disorders. This paper presents effectively one experiment with bulk RNA seq at 3 different timepoints, on a model previously published and characterised already, which focused on the expression of different transcripts that according to the authors indicate specific patterns of maturation for human NMJs.

While it is indeed true that human NMJs and dataset on their maturation and transcriptional (as well as functional profile) are not a widely available resource and need more development to be established, this reviewer is not convinced that the study in the present form constitute a significantly valuable and established (or useful) resource for this field, at least in its present form.

We thank the reviewer for highlighting that comprehensive datasets on human NMJ maturation are still rare and that further development is needed before such resources can be considered fully established. In the revised manuscript, we have tempered our original claim and now present our dataset as an initial temporal transcriptomic reference rather than a definitive or comprehensive resource. We also explicitly outline key limitations and opportunities for improvement, including functional validation, lineage-specific resolution, and integration with disease models in the discussion.

This entire study is one experiment, effectively, and does in no way advance or improve the published NMJ model itself. The data presented is entirely descriptive, which in itself wouldn't be a problem, but the author have provided no comparison with any other established model, in vivo direct systems, established murine NMJs in vivo or in vitro, or any other relevant comparison. Indeed, some of the observed pattern of expression suggest a potential coherence between the model and the in vivo hNMJ development (albeit, in the most vague and general of terms), but to prove that this system can be used as a valuable resource there need to be more validation.

We recognize that this work is primarily descriptive. Accordingly, we have revised the framing in the Introduction (p.2, lines. 29-30) and Discussion (p.3, lines 4-6) to clarify that our goal is to provide a temporal transcriptomic reference for human NMJ maturation, rather than introduce a novel experimental platform or extensive comparative analysis. We view this dataset as a resource to inform future mechanistic and disease-focused studies.

“Despite these advances, a systematic comparison of how closely each system reflects hNMJ morphology, function, and developmental timing is still lacking.”

“Together, these data demonstrate that human 3D neuromuscular co-cultures recapitulate transcriptional features of embryonic NMJ development and provide a temporal reference framework to inform future studies of NMJ formation and disease.”

The paper is also not helped by the choice of writing, which completely forgoes any explanation of the culture system, merely cites it, and presents Results and Discussion joined, which results in virtually no discussion at all on the potential pros and cons, considerations etc. on this system.

We thank the reviewer for this helpful comment on the manuscript structure and exposition. We have revised the manuscript to separate results and discussion into distinct sections, improving overall clarity. In addition, we have expanded the introductory (p.2, lines 31-34, lines 36-38) description of our culture system to ensure that key methodological considerations – including cell sourcing, differentiation timeline, and general features of the 3D neuromuscular co-culture – are clearly explained in the main text rather than only cited. We also now provide more explicit discussion of the relative strengths and limitations of this approach.

“In this study, we used a 3D human neuromuscular co-culture system that combines iPSC- derived spinal MNs with primary human myoblast-derived muscle fibers embedded in a fibrin-based hydrogel matrix within dumbbell-shaped PDMS molds. This configuration promotes alignment, myotube fusion, contractility, and NMJ formation. We selected this platform for its reproducibility, accessibility, and compatibility with live imaging and defined time molecular profiling.”

Moreover, the authors only use 2 ips lines for validation, and while the trends observed are similar there are clearly variations that would suggest at least 3-4 lines (as it is usually done in these studies) would have been ideal.

We appreciate the reviewer's comment regarding the number of iPSC lines used for validation. While we acknowledge that including additional lines (3-4) would further strengthen the generalizability of our findings, we now acknowledge this limitation in the Discussion (p.9, lines 12-15) and recommend expansion to additional genetic backgrounds in future work.

“We also acknowledge that the dataset presented here is derived from only two independent iPSC lines. Expanding future analyses to include additional donor genetic backgrounds will be important to capture inter-individual variability and to strengthen the generalizability of the resulting transcriptional reference.”

Our analysis using two independent hiPSC lines revealed highly comparable temporal transcriptional trajectories. As detailed in the revised Results section (Fig. 4B) (p.7, lines 36-39; p.8, lines 1-6), hierarchical clustering showed that samples primarily grouped by time point—indicating that developmental stage was the main driver of transcriptomic variation— while line-specific differences were modest and consistent. Importantly, both lines exhibited reproducible stage-specific transitions, without evidence of developmental lag or divergence. This supports the robustness of the co-culture model across genetically distinct lines.

“Hierarchical clustering of RNA-seq samples based on Euclidean distance showed that gene expression patterns were primarily organized by time points in culture rather than by iPSC line (Fig 4B). Samples collected at 1, 3, and 6 weeks each formed distinct clusters, highlighting developmental stage as the dominant factor shaping transcriptional profiles. As expected, week 1 samples from both iPSC#1 and iPSC#2 were grouped together and clearly separated from those at later stages. Week 3 and 6 samples also clustered by time point, with greater similarity between them compared to week 1, reflecting progressive maturation of the co-cultures. Within each time point, samples were further segregated according to iPSC line, suggesting a subtle but consistent line-specific transcriptional signature. This trend was most evident at 6 weeks, where iPSC#2 samples clustered more tightly, indicating higher within-line consistency.”

Second decision letter

MS ID#: bio.062196R1

MS TITLE: Temporal transcriptomic profiling of human 3D neuromuscular co-cultures

AUTHORS: Patrick Lüningschrör; Neha Jadhav Giridhar; Bitu Hambrecht; Maren Schenke; Bettina Seeger; Thorsten Bischler; Michael Briese

I am happy to tell you that your manuscript has been accepted for publication in Biology Open, pending our standard publication integrity checks. It was accepted on 6th August 2025.